# Histamine: A Bacterial Signal Molecule

**DOI:** 10.3390/ijms22126312

**Published:** 2021-06-12

**Authors:** Tino Krell, José A. Gavira, Félix Velando, Matilde Fernández, Amalia Roca, Elizabet Monteagudo-Cascales, Miguel A. Matilla

**Affiliations:** 1Department of Environmental Protection, Estación Experimental del Zaidín, Consejo Superior de Investigaciones Científicas, Prof. Albareda 1, 18008 Granada, Spain; felix.velando@eez.csic.es (F.V.); elizabet.monteagudo@eez.csic.es (E.M.-C.); 2Laboratory of Crystallographic Studies, IACT (CSIC-UGR), Avenida de las Palmeras 4, 18100 Armilla, Spain; jgavira@iact.ugr-csic.es; 3Department of Microbiology, Facultad de Farmacia, Campus Universitario de Cartuja, Universidad de Granada, 18071 Granada, Spain; matildefernandez@ugr.es (M.F.); amaliaroca@ugr.es (A.R.)

**Keywords:** histamine, signal molecule, sensing, *Pseudomonas aeruginosa*, histamine receptors, chemotaxis, gut microbiome

## Abstract

Bacteria have evolved sophisticated signaling mechanisms to coordinate interactions with organisms of other domains, such as plants, animals and human hosts. Several important signal molecules have been identified that are synthesized by members of different domains and that play important roles in inter-domain communication. In this article, we review recent data supporting that histamine is a signal molecule that may play an important role in inter-domain and inter-species communication. Histamine is a key signal molecule in humans, with multiple functions, such as being a neurotransmitter or modulator of immune responses. More recent studies have shown that bacteria have evolved different mechanisms to sense histamine or histamine metabolites. Histamine sensing in the human pathogen *Pseudomonas aeruginosa* was found to trigger chemoattraction to histamine and to regulate the expression of many virulence-related genes. Further studies have shown that many bacteria are able to synthesize and secrete histamine. The release of histamine by bacteria in the human gut was found to modulate the host immune responses and, at higher doses, to result in host pathologies. The elucidation of the role of histamine as an inter-domain signaling molecule is an emerging field of research and future investigation is required to assess its potential general nature.

## 1. Introduction

Bacteria have evolved a large number of signal transduction systems that recognize different signals and generate, in return, adaptive responses. Major protein families include transcriptional regulators, two-component systems (TCS), chemoreceptors, proteins involved in the synthesis and hydrolysis of the c-di-GMP and c-di-AMP second messengers, extracytoplasmic function (ECF) sigma factors and Ser/Thr/Tyr kinases [1]. Stimuli recognized are diverse and include an enormous variety of low molecular weight compounds, pH, temperature, light or osmotic stress, among others [2,3]. Major forms of signaling responses are transcriptional regulation, chemotactic movements or alterations in second messenger levels [1].

Frequently, bacteria establish interactions with organisms of other domains like animals, humans or plants. These interactions can either be of mutual benefit or part of a virulence strategy. A number of central signal molecules have been described that are synthesized and secreted by the bacterium as well as by its host. These signal molecules mediate intra- and inter-species communication that regulates multiple metabolic and physiological processes in both bacteria and their hosts [4,5]. The investigation of the role of these central signal molecules in the inter-domain crosstalk is a rapidly expanding field of research. A representative example for such central inter-species signals is the auxin indole-3-acetic acid (IAA) that is a key plant hormone, regulating, among other processes, plant growth and development [6]. However, IAA is commonly produced and secreted by bacteria that interact with plants and plays a key role in mediating plant–bacteria interactions [7,8]. Another example is the crosstalk that exists between the human gastrointestinal hormones, epinephrine and noradrenaline, and bacterial autoinducers to modulate bacterial physiology and metabolism, as well as the host’s inflammatory responses [9].

There is now emerging evidence, reviewed in this article, that histamine may be another central signal molecule that mediates bacteria–host interactions. Histamine is primarily known for its central role it plays in humans. It is a human neurotransmitter, a modulator of inflammatory reactions and the immune response and a key mediator of several events in allergies and autoimmune diseases [10]. Further activities of histamine include a participation in cell proliferation, differentiation, hematopoiesis, embryonic development, secretion of pituitary hormones as well as a regulation of gastrointestinal, cardiovascular and circulatory functions [11]. Histamine is synthesized from L-histidine by histidine decarboxylase (HDC). It is primarily secreted by mast cells and basophiles, and it exerts its function through four different types of histamine receptors, termed H1R, H2R, H3R and H4R [11].

A wide range of Gram-positive and Gram-negative bacteria were found to possess HDC-encoding genes and to synthesize histamine [12]. There appear to be two bacterial HDC superfamilies, namely those that require pyridoxal phosphate as a coenzyme, found primarily in Gram-negative bacteria, and those in Gram-positive species that employ a covalently bound pyruvate moiety for catalysis [12]. The regulation of the expression of *hdc* genes has been studied in several bacteria. Histidine was found to induce the expression of *hdc* genes [13,14,15,16], whereas histamine slightly repressed its expression in several lactic bacteria belonging to the *Lactobacillus, Pediococcus and Oenococcus* genera [14].

Apart from their capacity to synthesize histamine, more recent studies have shown that some bacteria are able to metabolize histamine. *Pseudomonas* species are characterized by an enormous metabolic versatility [17], and de la Torre et al. revealed that *P. putida* U is able to grow aerobically in a minimal medium, containing histamine as the sole carbon source [18]. The degradation of histamine coincided with the appearance of imidazole-4-acetic acid (ImAA), suggesting that the latter compound is a major intermediate in the degradation route. The authors showed that 11 proteins (HinABCDFLHGIJK), encoded in four different genomic regions (clusters *hin1* (*hinABCD*), *hin2* (*hinFLHG*) and *hin3* (*hinIJ*) and the stand-alone *hinK* gene), are required for histamine degradation in *P. putida* U [18]. Of these proteins, one is a histidine permease (HinA), three are transcriptional regulators (HinB, HinJ and HinK) and the remaining proteins are catabolic enzymes. A six-step catabolic process converts histamine into aspartic acid that is then converted into the tricarboxylic acid (TCA) cycle intermediate, fumaric acid [18]. To determine to which extent other bacteria may be able to degrade histamine, the authors inspected genomes for the presence of *hin* genes. These genes were commonly present in strains of the genus *Pseudomonas* but absent from any of the as-yet sequenced Gram-positive bacteria [18].

The detection of signal molecules by bacteria can serve several purposes: (i) they can indicate the presence of a compound of metabolic value or toxicity, or (ii) they can inform the bacterium of its present environmental niche. For a number of signal molecules, the physiological purposes of sensing appear to be tightly interwoven and include the metabolic aspect as well as the aspect of gaining information on the ecological niche. Histamine may be one of these signals. Bacterial histamine signaling is an emerging field of research that is reviewed here. In the first part of this article, we reviewed studies illustrating the histamine sensing capacity of bacteria, whereas we focused attention, in the second part, on the consequences of bacterial histamine secretion on the host.

## 2. Histamine Sensing by Bacteria

### 2.1. Pseudomonas aeruginosa PAO1

*P. aeruginosa* is among the most feared human pathogens. It is an opportunistic pathogen that infects virtually any tissue [19] and is the leading cause of nosocomial infections, particularly in immunocompromised, cancer, burn-wound and cystic fibrosis patients [20] and a frequent cause of bacteremia [21]. The World Health Organization (WHO) has placed *P. aeruginosa* second on the global priority list of antibiotic-resistant bacteria to guide research, discovery and development of new antibiotics and has rated the development of new antimicrobial agents against *P. aeruginosa* as critical [22]. Strain PAO1 was found to be able to grow on histamine as a sole carbon and nitrogen source, indicating that it harbors a functional histamine degradation pathway [23]. In a subsequent study, it was found that the histamine catabolic pathway described in the non-pathogenic *P. putida* U [18] is also highly conserved in the opportunistic human pathogen *Pseudomonas aeruginosa* (Figure 1) [24].

#### 2.1.1. Transcriptional Responses to Histamine Exposure

The effect of histamine on bacterial gene transcript levels was assessed, for the first time, using *P. aeruginosa* PAO1 as a model system [24], the primary reference strain for this pathogen. RNA-seq experiments were conducted, comparing the wild-type (wt) strain in the absence and presence of 2 mM histamine, and samples were taken 3 h after histamine addition. This study showed that the transcript levels of approximately 8.5% of the PAO1 genes showed at least a three-fold change. There were, in total, 301 upregulated and 178 downregulated genes, a selection of which is shown in Table 1.

The authors selected nine genes and determined histamine-induced changes in transcript levels using quantitative real-time PCR (RT-qPCR), and the results obtained were consistent with RNA-seq data. Furthermore, additional RT-qPCR studies showed significant changes in the transcript levels of the *hinD*, *hinF*, *pvdS* and *pqsA* genes at a 1000-fold lower histamine concentration (2 µM), indicative of high-affinity signal recognition [24].

Histamine exposure caused significant changes in histamine-related genes, such as genes encoding enzymes for histamine metabolism (HinCDFLHG), transport (HinA) and regulation (HinK) (Table 1, Figure 1).

Of note are the large changes that have been observed for the genes involved in histamine metabolism that ranged from a 240- to 2200-fold increase in the presence of histamine [24]. Many of the identified histamine-regulated genes were either directly or indirectly related to different virulence processes (Table 1). A large number of the upregulated genes were associated with iron uptake, such as those encoding proteins for the synthesis and secretion of the pyoverdin and pyochelin siderophores, iron transport or different ECF sigma factors (Table 1). Further upregulated genes encoded proteins involved in the synthesis of the *Pseudomonas* quinolone quorum sensing signal (PQS) or the type III secretion system. Another group of upregulated genes had regulatory functions, such as the transcriptional regulators ToxR, PtrB, MvaT, VqsM, PsrA and RsaL (Table 1) that regulate diverse processes, such as the expression of genes encoding the primary toxin endotoxin A [36], quorum sensing proteins [39] or the type III secretion system [41]. Alternatively, several of the downregulated genes also had regulatory functions (Table 1) [24], such as components of the Che2 chemosensory pathway, which is of unknown function but related to virulence [46,47], or the quorum-sensing PprAB two-component system [48]. In addition, two chemotaxis chemoreceptors were downregulated, including CtpH, a chemoreceptor specific for inorganic phosphate, a major signal regulating *P. aeruginosa* virulence [51]. 

#### 2.1.2. A Large Part of Histamine-Dependent Transcriptional Responses Are Mediated by the Transcriptional Regulator HinK

Among the genes that were upregulated in the presence of histamine was *hinK*, encoding a LysR-type transcriptional regulator. HinK was first identified in *P. putida* U, and it regulates histamine catabolism in this strain, together with the transcriptional regulators HinB and HinJ, as described above [18]. In PAO1, the *hinK* gene was found to be next to the *hinDAC* genes that were involved in histamine metabolism and transport (Figure 1). To assess the role of HinK in the histidine-mediated regulation, the authors conducted RNA-seq experiments, comparing the wt with the *hinK* mutant in the presence of histamine, showing a significantly changed pattern in the gene transcript levels with respect to the experiment comparing the histamine-free and -supplemented wt strain [24]. To verify whether HinK controls the expression of the *hinDAC* genes, the authors constructed a *hinD* promoter-*lux* transcriptional fusion. In the wt strain, the addition of histamine caused an important increase in the transcriptional activity, whereas no changes were observed in the *hinK* mutant, a phenotype reversed by mutant complementation [24]. Analogous experiments showed that HinK also regulates the transcription of the *hinFLHG* operon (Figure 1) as well as its own expression [24]. Electrophoretic mobility shifts revealed that HinK binds to the *hinD* and *hinF* promoters, and a conserved sequence motif in both promoters was identified as the HinK operator site [24].

HinK is composed of a DNA binding- and ligand-binding domain (LBD), and experiments were conducted to identify the signal that binds and activates HinK. Several pieces of evidence indicate that HinK does not recognize histamine directly but instead imidazole-4-acetic acid (ImAA), which corresponds to an intermediate in the metabolic pathway converting histamine into aspartic acid, as described above (Figure 1B). Electrophoretic mobility shift assays revealed that micromolar concentrations of ImAA caused HinK binding at its target DNA, namely *hinD* and *hinF* promoters, an observation that was not made using a variety of related compounds, including histamine [24]. These data are not fully consistent with the relatively low affinity, of 1.56 mM, for the binding of ImAA to HinK [24]. The authors reported the three-dimensional structure of the apo HinK protein and ImAA binding studies to site-directed HinK mutants indicate that the ligand binds between both lobes of the LBD in a manner similar to other LysR type transcriptional regulators [24].

#### 2.1.3. HinA Is a Histamine Transporter Permitting Histamine Uptake and Sensing by HinK

HinA is an APC (amino-acid-polyamine-organocation)-type transporter, and several pieces of evidence indicate that it is the primary histamine transporter. In *P. putida* U, a mutant in the *hinA* gene was unable to take up tritium-labelled histamine, a phenotype that was reversed by complementation with the *hinA* gene [18]. Wang et al. refered to *P. aeruginosa* PA0220 as the HinA homologue [24]; however, the sequence identity between both proteins, with 17%, is very modest. They showed that the deletion of *P. aeruginosa hinA* significantly reduced the transcriptional activity from the *hinD* promoter that was found to be controlled by HinK in response to histamine, which supports the notion that HinA is the primary histamine transporter. Transporters often employ extracytosolic solute binding proteins that present the transport substrate to the permease [54]. In close vicinity to the *hinA* gene is a gene encoding a solute-binding protein, PA0222 (Figure 1A), and histamine was found to increase its transcript levels by 400-fold (Table 1). However, microcalorimetric titrations of purified PA0222 showed that it bound γ-aminobutyrate with nanomolar affinity but failed to recognize histamine [55]. The potential role of PA0222 in histamine transport is thus unclear.

#### 2.1.4. Histamine and HinK Regulate *P. aeruginosa* Virulence

Based on the observation that histamine induces the expression of many virulence-related genes, Wang et al. conducted experiments to elucidate, in more detail, the role of histamine in *P. aeruginosa* virulence [24]. Using the *Drosophila melanogaster* model, the authors showed that histamine treatment increased bacterial virulence, whereas no change in virulence was noted for the *hinK* mutant, a phenotype that was reversed by complementation with *hinK*. The same strains were analyzed in a mouse acute lung infection model. In accordance with the above data, the deletion of *hinK* caused a significant reduction in virulence as compared to the wt and the complemented mutant strain [24]. The data thus indicate that histamine is a signal molecule that regulates *P. aeruginosa* virulence.

#### 2.1.5. Histamine Chemotaxis

*P. aeruginosa* PAO1 and *P. putida* KT2440 were found to move chemotactically to histamine [23]. The onset of chemotaxis occurred for *P. aeruginosa* at the unusually low concentration of 500 nM, whereas initial responses of *P. putida* were observed at 5 µM. Over the entire histamine concentration range tested, i.e., 500 nM to 50 mM, the magnitude of chemotaxis of PAO1 was well superior to that of KT2440 [23]. Maximal responses of PAO1 were detected at 5 mM. Strain PAO1 has 26 chemoreceptors, of which 23 were predicted to stimulate the chemotaxis pathway [46,56]. Experimentation with a number of chemoreceptor mutants revealed that the histamine chemotaxis was not based on a single chemoreceptor, like for many other chemoeffectors studied, but on the concerted action of the TlpQ, PctA and PctC chemoreceptors [23]. Interestingly, mutants in *pctA* and *pctC* failed to respond to a high histamine concentration (i.e., 5–50 mM), whereas the *tlpQ* mutant did not respond to low concentrations (i.e., 500 nM–500 µM) [23]. Therefore, the combined action of three chemoreceptors with different sensitivities broadened the response range, a finding reminiscent of the action of the CtpL and CtpH chemoreceptors for inorganic phosphate [50,57]. Like histamine, inorganic phosphate is of central physiological relevance, since it is a key signal that regulates the expression of many virulence-related genes [51,58]. It is tempting to speculate that the recognition of a specific signal molecule by multiple chemoreceptors reflects a particular physiological relevance of the signal. PctA and PctC have previously been shown to bind and mediate chemoattraction to different proteinogenic amino acids and γ-aminobutyrate [59,60,61]. Both receptors possess a dCache type LBD [62] that binds proteinogenic amino acids and γ-aminobutyrate directly [59,60]. However, microcalorimetric titrations of the individual PctA and PctC LBDs with histamine did not show binding [23]. It was thus suggested that histamine recognition by both receptors occurs via the binding of solute-binding proteins [23], an indirect mechanism for the activation of different bacterial sensor proteins that appears to be widespread among bacteria [63].

#### 2.1.6. The Chemoreceptor TlpQ Binds Histamine at its Ligand-Binding Domain with High Affinity

In contrast to PctA and PctC, the LBD of the TlpQ chemoreceptor bound histamine directly [23]. Microcalorimetric titrations of the TlpQ sensor domain revealed a dissociation constant of 0.64 µM that corresponded to an affinity significantly higher than the average for ligand recognition by chemoreceptor LBDs [64]. In addition to histamine, TlpQ also recognized structurally related polyamines, namely putrescine, cadaverine, spermidine, agmatine and ethylenediamine, with a similarly high affinity [23]. As stated above, the magnitude of histamine chemotaxis in *P. putida* KT2440 was inferior to that of *P. aeruginosa* PAO1. This finding may be related to the fact that the LBD of the McpU chemoreceptor, the TlpQ homologue in KT2440 [65], recognizes histamine with a 40-fold lower affinity [23].

Like PctA and PctC, the TlpQ chemoreceptor has a dCache type LBD, and its 3D structure in a complex with histamine has been solved by X-ray crystallography [23] (Figure 2). The TlpQ-LBD is composed of two structural α/β modules, and histamine was bound at the membrane distal module, like in the very large majority of other characterized dCache domains [59,65,66,67,68]. The molecular detail of histamine recognition by human receptors has recently been deciphered by reporting three dimensional structures of the Histamine H1 receptor [69] and the β3 GABA_A_ receptor in a complex with histamine [70] (Figure 2). 

The comparison of TlpQ-LBD with the two human receptors thus shows that the proteins involved in histamine sensing in bacteria and humans are entirely different. In the human H1 receptor, histamine is recognized within the membrane by several transmembrane helices, whereas histamine is bound to the extracytosolic part of the β3 GABA_A_ receptor, where it is recognized by a curved β-sheet. Although the 3D structures of the three histamine receptors are entirely different, there was a certain parallelism between TlpQ and the β3 GABA_A_ receptor in the molecular detail of ligand recognition that is illustrated in Figure 3. 

In both cases, the primary and secondary histamine amino groups are coordinated by negatively charged amino acids and a series of tyrosine residues that interact with the linear and cyclic parts of histamine. 

### 2.2. Escherichia coli

Whereas a significant part of the transcriptional responses in *P. aeruginosa* appear to be related to the sensing of a histamine metabolite by the HinK transcriptional regulator, the two-component system AtoSC appears to be involved in histamine sensing in *E. coli*. Inspection of the sequence of the AtoS sensor kinase in Pfam [72] indicated that it has two transmembrane regions that flank a potential periplasmic sensor domain that is un-annotated, but homology modeling using Phyre2 [73] indicated that it is likely to form an α/β fold, similar to an sCache domain. In addition, AtoS has a cytosolic PAS domain that may also be involved in signal sensing. On the other hand, the AtoC response regulator is a member of the NtrC-NifA family of transcriptional regulators and is composed of an N-terminal receiver domain, followed by an AAA+_ATPase and DNA-binding domain [74]. AtoSC is encoded upstream of the *atoDAEB* gene cluster that encodes proteins involved in the catabolism of short chain fatty acids (SCFAs) [75], and AtoSC was found to control the expression of this operon [76,77]. SCFAs are important signal molecules in the human gut microbiome. They are produced in the colon following microbial fermentation of dietary fibers, are important energy sources for colonocytes and regulate the assembly and organization of tight junctions [78]. Abnormalities in SCFA levels, either caused by dysbiosis (i.e., alteration of gut microbiota homeostasis) or diet, were suggested to play a role in a number of pathologies, such as type-2 diabetes, obesity, inflammatory bowel disease, colorectal cancer or allergy [79].

The direct action of the AtoSC TCS on the expression of the *atoDAEB* operon also modulated the synthesis of the complexed poly-(R)-3-hydroxybutyrate (cPHB), a ubiquitous cell compound that contributes to Ca^2+^ homeostasis [80]. In addition, AtoSC also contributes to the regulation of flagellar gene expression and was thus shown to modulate motility and chemotaxis [81]. Spermidine and acetoacetate are the effectors of the AtoSC system [82,83]. Multiple pieces of evidence have suggested that AtoSC activity is modulated by Ca^2+^ that may act as a co-signal [84,85]. However, the molecular detail and the corresponding sensor domains of these effectors have so far not been established.

Evidence has been presented showing that histamine interferes with AtoSC activity. Histamine was shown to increase *atoC* transcription and to reduce cPHB biosynthesis [84,86]. cPHB biosynthesis requires SCFAs [83], and the interference of histamine with SCFA metabolism and levels may play a regulatory role in the gut. Furthermore, low concentrations of histamine enhanced motility and chemotaxis in *E. coli*, whereas the opposite effect was noted when histamine was present at higher levels [81]. This histamine-mediated regulatory effect was not observed in a strain that contained a truncated version of AtoC that lacked the receiver domain [81]. However, the molecular mechanism by which histamine modulates AtoSC function remains unknown.

## 3. Histamine Release by Bacteria and Its Consequences

Apart from the fact that bacteria sense histamine, there is evidence that bacterial-derived histamine has multiple consequences, for example, on host health [78] and food safety [87]. In fact, histamine levels are monitored in a number of different foods as a measure of food freshness [88,89]. This is particularly relevant for seafood products, where bacteria-secreted histamine can provoke food poisoning [87]. The list of microorganisms that secrete histamine in seafood is long and includes Gram-positive and Gram-negative species. Most abundant are *Enterobacteriaceae* belonging to genera such as *Morganella*, *Enterobacter*, *Hafnia*, *Proteus* and *Photobacterium*, as well as different pseudomonads and lactic acid bacteria of the genera *Lactobacillus* and *Enterococcus* [87].

The effect of histamine secretion by human intestinal bacteria on its host is a more recent but rapidly expanding field of research. Initial in vitro studies showed that histamine suppressed the chemokine and proinflamatory cytokine secretion in human monocyte-derived dendritic cells [90]. Murine studies showed that the administration of the histamine-secreting *Lactobacillus rhamnosus* had an anti-inflammatory effect, as evidenced by a reduction in the secretion of various interleukines and tumor necrosis factor α. This effect was lost in animals deficient in the histamine 2 receptor, indicating that microbiota-derived histamine could be immunomodulatory [90]. Administration of another *Lactobacillus* species, *L. saerimneri,* that is able to secrete approximately 100-fold more histamine as compared to *L. rhamnosus*, resulted, next to a variety of immune responses, in animal weight loss and signs of deteriorating health [91]. The authors suggested that the amount of histamine secreted by a microbe may be critical in determining the nature of the effect.

Another study reported that the abundance of histamine-secreting bacteria is increased in the gut of adult asthma patients. This study thus challenged the widespread concept that human mast cells and basophiles are the principal histamine sources. These data thus also suggest that bacterial-derived histamine contributes to histamine-mediated pathologies [92]. In addition, data indicting another link between bacterial-derived histamine and pathology was reported by Gallardo et al. [93]. The authors compared gut microbiota composition and metabolome in stool samples obtained from healthy children and children with diarrhea positive for diarrheagenic *E. coli* (DEC). Metabolomic studies revealed higher histamine concentrations in the DEC group as compared to healthy children, and altered histamine levels were associated to certain gut microbiota species such as *Enterobacter hormaechei*, *Bifidobacterium stercoris* and *Shigella* spp. [93]. More recent studies have suggested that bacterial histamine release in the gut does not only cause a local modulation of the host immune system, but can also have immunological consequences at distant mucosal sites, such as in the lung [94]. *E. coli* was engineered to secrete histamine and administered orally to mice [94]. The authors observed an anti-inflammatory response in the lung, as evidenced by reduced inflammatory cell numbers in bronchoalveolar lavages. Experimentation with mice deficient in the histamine 2 receptor (H2R) showed that the anti-inflammatory effect of bacterial-derived histamine is partially mediated by this receptor [94]. During the investigation of the impact of different metabolites produced by gut bacteria on host physiology, the effect of bacteria-produced histamine was evaluated [95]. Gut bacteria of the *Morganella morganii* and *Lactobacillus reuteri* species were found to produce histamine in vivo during the colonization of the mouse intestine, and L-His dietary supplementation increased histamine production by these bacteria. In this study, the authors found that bacteria-derived histamine was associated with increased mice colon motility and fecal output and that treatment with histamine receptor antagonists largely blocked the effect of bacterial histamine on colon motility [95].

Irritable bowel syndrome (IBS) is a common gastrointestinal disorder, and accumulating evidences at both preclinical and clinical levels indicates an involvement of enteric microbiota in its pathogenesis [96]. Histamine levels and the abundance of *hdc* genes was determined in both healthy and IBS patients using metabolomics and metagenomics data from the integrative Human Microbiome Project. These analyses revealed that IBS patients presented higher levels of histamine and bacterial *hdc* genes [95]. Subsequent studies also showed that supernatants from colonic samples of IBS patients contained increased histamine levels, and expression levels of the histamine receptors H1R and H2R were upregulated in IBS patients [97]. The authors thus hypothesized that a dysbiosis with increased histamine-secreting or HDC-containing bacteria was potentially associated with the development and aggravation of IBS [96].

## 4. Outlook

The knowledge available on the role of histamine as a bacterial signal molecule is summarized in Figure 4.

The elucidation of the role of histamine as a signal molecule for inter-domain communication is an emerging field of research that requires future efforts. So far, the information on histamine sensing is restricted to *P. aeruginosa, P. putida* and *E. coli*, and studies need to be conducted to determine to which extent other species show similar responses. The main physiological roles of chemotaxis are gaining access to compounds that serve for growth, the perception of information on the environmental niche or the localization of sites that are suitable for attachment or invasion. For key signal molecules that are of metabolic value, like histamine, it has to be determined whether the primary motivation of chemotaxis is related to metabolism or the capacity to infect hosts. There is initial evidence that bacterial histamine secretion in the gut microbiome is associated with digestive disorders. In this context, important gaps in knowledge to be closed are the determination of environmental factors that may trigger histamine release and to determine the capacity of histamine release for strains typically found in the gut microbiome. Such information would facilitate the diagnosis of histamine-related disorders from the composition of the patients’ microbiome.

## Figures and Tables

**Figure 1 ijms-22-06312-f001:**
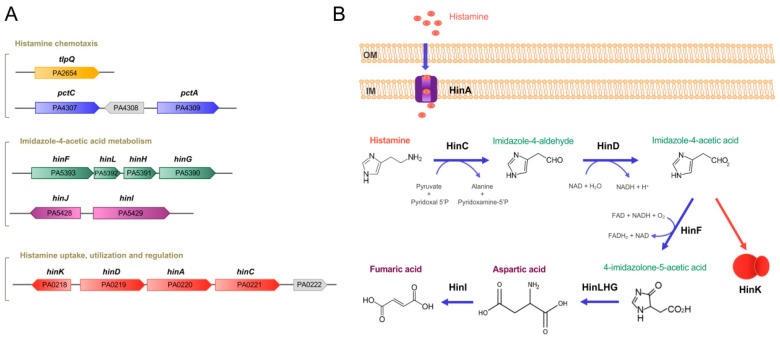
Genes and proteins involved in histamine metabolism, transport, regulation and chemotaxis in *P. aeruginosa* PAO1. (**A**) Genetic organization of genes. (**B**) The proposed histamine catabolic pathway. Data are based on [18,23,24].

**Figure 2 ijms-22-06312-f002:**
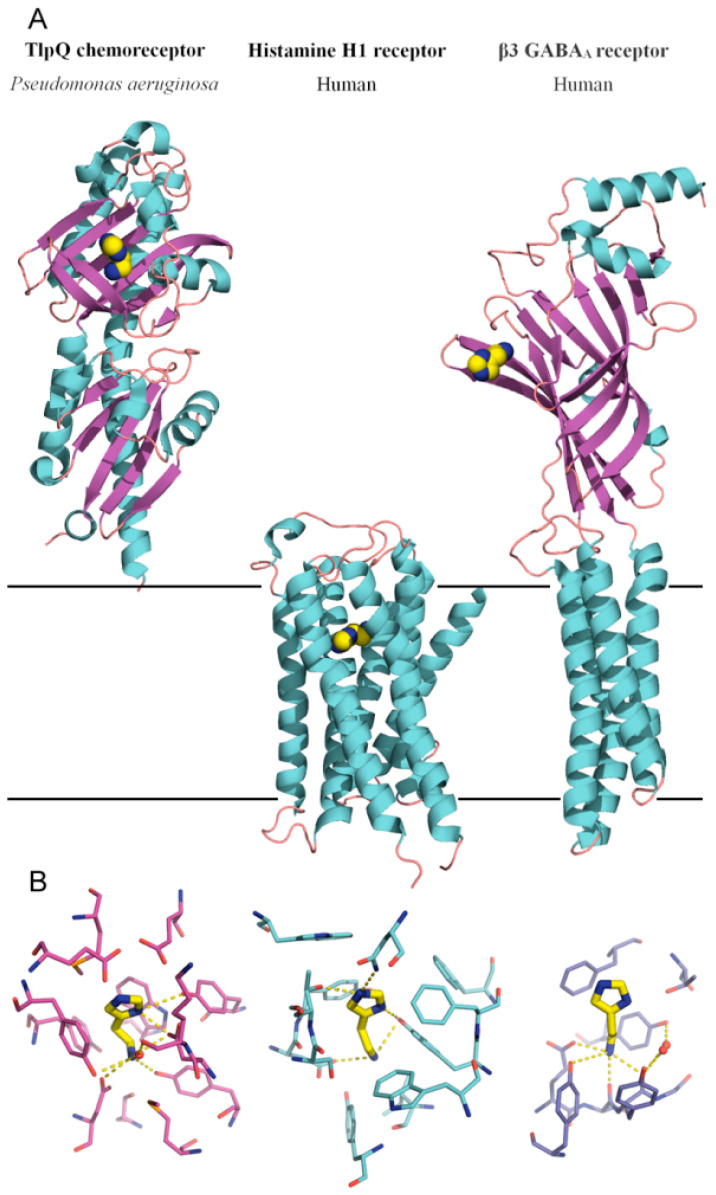
Bacterial and human histamine receptors. (**A**) Shown is the ligand-binding domain of the TlpQ chemoreceptor from *P. aeruginosa* PAO1 (PDB ID 6FU4), the human histamine H1 receptor (PDB ID 7DFL) and the human β3 GABA_A_ receptor (PDB ID 7A5V). Bound histamine is shown in stick mode in the lower part of the figure. These structures have been published in [23,69,70]. (**B**) Zoom on the histamine binding sites of the receptors shown above.

**Figure 3 ijms-22-06312-f003:**
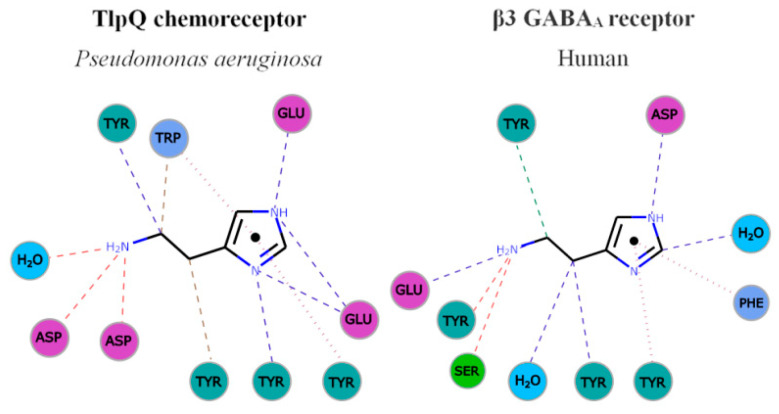
Parallelism in the mode of histamine recognition by *P. aeruginosa* TlpQ-LBD and the human β3 GABA_A_ receptor. The interaction of histamine within the different binding pocket was automatically generated at the PDBe, using Arpeggio [71]. Non-covalent interactions are shown by the following colored dashed lines: red, hydrogen bonds; green, hydrophobic interactions; brown, weak hydrogen bonds; and purple, pi-pi interactions. The thickness of each dash is related to the interaction-distance. Hydrophobic, negatively charged, aromatic and polar residues are colored in blue, magenta, green and cyan, respectively. For clarity, only some representative interactions are shown.

**Figure 4 ijms-22-06312-f004:**
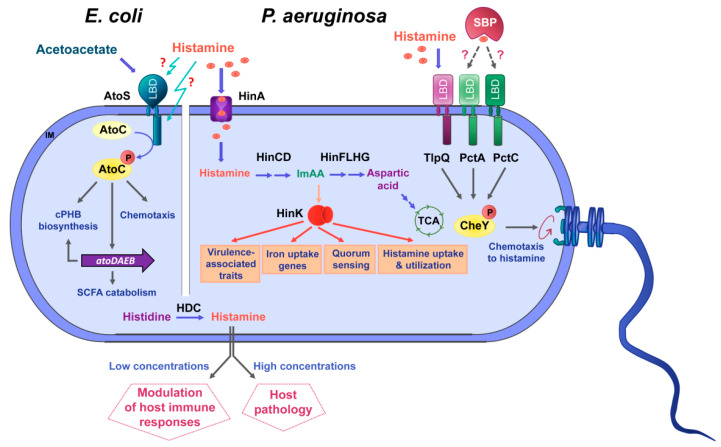
Summary of data available on histamine sensing and secretion by bacteria. On the left, histamine sensing by the TCS AtoSC in *E. coli*. On the right, histamine assimilation and chemotaxis in *P. aeruginosa*. Lower part: Many bacteria synthesize histamine by a decarboxylation of histidine using the histidine decarboxylase (HDC) and secrete histamine. Blue arrows: metabolic pathways; orange arrows: gene expression regulation; grey arrows: activation of biological processes; dotted lines: hypothetical interaction; LBD: ligand-binding domain; SBP: solute-binding protein; ImAA: imidazole-4-acetic acid; HinA: permease for the histamine uptake; HinCD: enzymes for the conversion of histamine to ImAA; HinFLHG: enzymes for the conversion of ImAA to aspartic acid; HinK: LysR-family response regulator; IM: inner membrane.

**Table 1 ijms-22-06312-t001:** The effect of histamine on *P. aeruginosa* PAO1 transcript levels. Shown is a selection of genes with altered transcript levels in an RNA-seq study comparing the wt strain in the absence and presence of 2 mM histamine. Many of these genes play a role in virulence. In total, approximately 8.5% of the *P. aeruginosa* genes showed at least a three-fold change. Data were taken from [24].

Gene ID	Name	Log_2_ Fold Change	Description	Function/ Comment	Ref. Function
**Histamine-mediated upregulation**					
**Histamine metabolism, transport and regulation**					
PA5390	*hinG*	7.9	Probable peptidic bond hydrolase	Histamine utilization	[24]
PA5391	*hinH*	10.9	Hypothetical protein		
PA5392	*hinL*	10.7	Conserved hypothetical protein		
PA5393	*hinF*	11.1	Conserved hypothetical protein		
PA0219	*hinD*	10.0	Probable aldehyde dehydrogenase		
PA0221	*hinC*	10.0	Probable aminotransferase		
PA0220	*hinA*	9.5	Histamine transporter	Histamine transport	
PA0218	*hinK*	4.8	Transcriptional regulator	Histamine-mediated regulation	
PA0222		8.7	Solute-binding protein	Possibly transport	
**Iron acquisition**					
PA0931	*pirA*	3.2	Ferric enterobactin receptor PirA		[25]
PA2385	*pvdQ*	6.3	3-oxo-C12-homoserine lactone acylase PvdQ	Siderophore pyoverdin synthesis, secretion, regulation and pyoverdin-Fe uptake	[26]
PA2386	*pvdA*	7.5	L-ornithine N5-oxygenase		
PA2389	*pvdR*	2.6	PvdR		
PA2390	*pvdT*	2.4	PvdT		
PA2392	*pvdP*	4.1	PvdP		
PA2394	*pvdN*	5.9	PvdN		
PA2395	*pvdO*	6.3	PvdO		
PA2396	*pvdF*	3.4	Pyoverdine synthetase F		
PA2397	*pvdE*	6.3	Pyoverdine biosynthesis protein PvdE		
PA2398	*fpvA*	6.0	Ferripyoverdine receptor		
PA2399	*pvdD*	2.9	Pyoverdine synthetase D		
PA2400	*pvdJ*	3.0	PvdJ		
PA2413	*pvdH*	5.6	L-2,4-diaminobutyrate:2-ketoglutarate 4-aminotransferase		
PA2424	*pvdL*	5.8	PvdL		
PA2425	*pvdG*	6.2	PvdG		
PA2426	*pvdS*	5.7	Sigma factor PvdS		
PA0472	*fiuI*	3.1	ECF sigma factor FiuI	Ferrichrome activated	[27]
PA2468	*foxI*	2.5	ECF sigma factor FoxI	Ferrioxamine activated	[28]
PA3410	*hasI*	2.9	ECF sigma factor HasI	Heme activated	[29]
PA4168	*fpvB*	3.3	Second ferric pyoverdine receptor FpvB	Pyoverdine transport	[30]
PA4221	*fptA*	1.7	Fe(III)-pyochelin outer membrane receptor precursor	Siderophore pyochelin synthesis and transport	[31]
PA4226	*pchE*	3.1	Dihydroaeruginoic acid synthetase		
PA4228	*pchD*	4.1	Pyochelin biosynthesis protein PchD		
PA4229	*pchC*	3.6	Pyochelin biosynthetic protein PchC		
PA4230	*pchB*	2.7	Salicylate biosynthesis protein PchB		
PA4231	*pchA*	2.3	Salicylate biosynthesis isochorismate synthase		
PA4687	*hitA*	3.3	Ferric iron-binding periplasmic protein HitA	Iron transport	[32]
PA4688	*hitB*	3.2	Iron (III)-transport system permease HitB		
**Quorum sensing**					
PA0996	*pqsA*	3.4	Probable coenzyme A ligase	*Pseudomonas* quinolone signal (PQS) quorum sensing system	[33]
PA0997	*pqsB*	3.8	PqsB		
PA0998	*pqsC*	3.8	PqsC		
PA0999	*pqsD*	3.8	3-oxoacyl-[acyl-carrier-protein] synthase III		
PA1000	*pqsE*	3.6	Quinolone signal response protein		
PA1001	*phnA*	3.5	Anthranilate synthase components I and II (important for PQS synthesis)	PQS synthesis	[34]
PA1002	*phnB*	3.0			
**Secretion system**					
PA1718	*pscE*	2.3	Type III export protein PscE	Type III secretion apparatus	[35]
PA1721	*pscH*	1.9	Type III export protein PscH		
PA1715	*pscB*	1.8	Type III export apparatus protein		
**Regulation**					
PA0707	*toxR*	1.9	Transcriptional regulator ToxR	Exotoxin A expression	[36]
PA0612	*ptrB*	2.0	Repressor PtrB	Type III secretion system expression	[37]
PA1431	*rsaL*	2.0	Regulatory protein RsaL	Virulence and biofilm formation	[38]
PA2227	*vqsM*	2.4	Transcriptional regulator VqsM	Quorum sensing and virulence	[39]
PA2686	*pfeR*	3.1	PfeR response regulator	Enterobactin receptor	[40]
PA2687	*pfeS*	2.6	PfeS sensor kinase		
PA3006	*psrA*	1.8	Transcriptional regulator PsrA	Type III secretion system	[41]
PA4315	*mvaT*	2.3	Transcriptional regulator MvaT	Type III secretion system	[42]
PA5124	*ntrB*	4.0	NtrB kinase	Invasiveness and Virulence	[43]
PA5125	*ntrC*	3.7	NtrC response regulator		
**Others**					
PA4760	*dnaJ*	3.2	Heat shock protein	Pyocyanin production	[44]
PA4761	*dnaK*	3.7	Chaperone DnaK	Translocation across the intestinal epithelia cells	[45]
**Histamine-mediated downregulation**					
**Regulation**					
PA0173	*cheB2*	−2.1	CheB_2_ methylesterase	Che2 pathway, unknown function, involved in virulence	[46,47]
PA0174	*cheD*	−2.2	CheD deamidase		
PA0175	*cheR2*	−2.4	CheR2 methyltransferase		
PA0176	*mcpB/aer2*	−2.3	Aer2/McpB chemoreceptor		
PA0177	*cheW*	−1.9	CheW coupling protein		
PA4293	*pprA*	−2.5	Sensor kinase PprA	Quorum sensingregulation	[48]
PA4296	*pprB*	−1.6	Response regulator PprB		
**Motility**					
PA1930	*mcpS*	−2.1	Chemoreceptor McpS	Chemotaxis	[49]
PA2561	*ctpH*	−2.4	Inorganic phosphate (Pi) specific chemoreceptor CtpH	Pi is a major virulence signal	[50,51]
**Others**					
PA4299-4306	*Flp-tad-rcp* locus	−2.3 to −4.8	Formation of type IVb pili	Aggregation and biofilm formation	[52]
PA4236	*katA*	−2.1	Major catalase KatA	Osmoprotection and virulence	[53]

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
