# Peer review of "Histamine: A Bacterial Signal Molecule"

_ijms, 2021, doi:10.3390/ijms22126312_

Round 1
Reviewer 1 Report
The manuscript describes the role of histamine in bacterial metabolism and signaling. Data is presented clearly and conclusions are well supported. There are, however, some minor points.
- Line 58: should be "plays in humans"
- Line 68: should be "EDC-encoding"
- Line 157: should be "it regulates"
- Line 279: should be "In both cases, "
- Figure 2. The data convincing but I am wondering if it is the only known mode of histamine binding. Could the Author comment on this?
Author Response
The manuscript describes the role of histamine in bacterial metabolism and signaling. Data is presented clearly and conclusions are well supported. There are, however, some minor points.
Response: We would like to thank this reviewer for his/her effort to review this manuscript and to suggest these changes.
- Line 58: should be "plays in humans"
Response: Thanks. Done.
- Line 68: should be "EDC-encoding"
Response: Thanks. Done.
- Line 157: should be "it regulates"
Response: Thanks. Done.
- Line 279: should be "In both cases, "
Response: Thanks. Done.
- Figure 2. The data convincing but I am wondering if it is the only known mode of histamine binding. Could the Author comment on this?
Response: The three structures shown are the result of a search in the protein database inspecting all structures that contain bound histamine. Within these structures, there were three histamine receptors, which are the ones shown in Fig. 2. However, in eukaryotic hosts there are three other histamine receptors (H2, H3 and H4) for which there is biochemical but no structural information on ligand binding. However, this information has not been reviewed since it would divert from the central theme of this review that consists in illustrating the relevance of histamine as bacterial signal molecule.
Reviewer 2 Report
Krell and colleagues present a comprehensive review on histamine as a signal molecule in bacteria and in relationship with the host. Most of the information provided, are on Pseudomonas aeruginosa, Pseudomonas putida and Escherichia coli for which detailed analyses have been carried out. Moreover, the authors present data on the role of secreted bacterial histamine in the health of the host.
The review is well structured and written. In some cases, it is too detailed but overall, it has a good flow. I don’t have general concerns but only minor comments which should be addressed in the final version.
Line 73: the or its should be removed
Line 121: what is the rationale of the selection of those genes? Is there an enrichment or where handpicked? In general, Table 1 is a repetition, considering that the categories of genes differentially expressed are already presented in the text. Moreover, the full data are also available in the specific publication.
Line 125-127: it is already generally recognized the robustness of RNA-seq. So there is no need to confirm the results by qRT-PCR.
Line 127-129: maybe the authors could comment of the difference, if any, in the differentially expressed genes at high and low histamine concentration.
Line 160-163: this is a repetition since the same article is presented above in section 2.1.1.
Is it known If P. aeruginosa can secrete histamine or if histamine can modulate increased virulence during exacerbation in the lungs of CF patients? Moreover, considering P. aeruginosa as an opportunistic pathogen, where in the host histamine is secreted which might attract and modulate its virulence? Maybe a subsection of paragraph 3 could be included presenting where histamine has been identified in the host.
The authors might include a figure similar to Figure 1 specific for E. coli.
Author Response
Krell and colleagues present a comprehensive review on histamine as a signal molecule in bacteria and in relationship with the host. Most of the information provided, are on Pseudomonas aeruginosa, Pseudomonas putida and Escherichia coli for which detailed analyses have been carried out. Moreover, the authors present data on the role of secreted bacterial histamine in the health of the host.
The review is well structured and written. In some cases, it is too detailed but overall, it has a good flow. I don’t have general concerns but only minor comments which should be addressed in the final version.
Response: We would like to thank this reviewer for his/her time and effort to review this manuscript. His/her comments are well appreciated.
Line 73: the or its should be removed
Response: Thanks, well spotted. The “the” has been removed.
Line 121: what is the rationale of the selection of those genes? Is there an enrichment or where handpicked? In general, Table 1 is a repetition, considering that the categories of genes differentially expressed are already presented in the text. Moreover, the full data are also available in the specific publication.
Response: We see the point of this reviewer. The rational for selecting these genes is that for almost all of them a role in virulence has been demonstrated in previous studies. In the revised version of this manuscript we have added the sentence “Many of these genes play a role in virulence.” to the legend of this Table. We are aware that these genes have been reported in a previous study. However, the authors of this study (Wang et al. (2021) Science Bulletin 2021, 66, 1101-1118) dedicated very little space in their publication on these RNA seq data. In our opinion these data are very important and should have merited more attention in the original publication, such as for example a Table similar to the one we include in our manuscript to convey the important message that histamine regulates the transcript level of many virulence-related genes.
Line 125-127: it is already generally recognized the robustness of RNA-seq. So there is no need to confirm the results by qRT-PCR.
Response: We agree. However, we cite the authors of this study (Wang et al. (2021) Science Bulletin 2021, 66, 1101-1118) who have stated that qRT-PCR experiments were conducted to verify the RNA-seq data.
Line 127-129: maybe the authors could comment of the difference, if any, in the differentially expressed genes at high and low histamine concentration.
Response: Thanks. We agree. The sentence “Furthermore, additional RT-qPCR studies showed significant changes at a 1000-fold lower histamine concentration (2 µM), indicative of high-affinity signal recognition [24].” has been replaced by “Furthermore, additional RT-qPCR studies showed significant changes in the transcript levels of the hinD, hinF, pvdS and pqsA genes at a 1000-fold lower histamine concentration (2 µM), indicative of high-affinity signal recognition [24].”
Line 160-163: this is a repetition since the same article is presented above in section 2.1.1.
Response: We beg to differ in this issue. In section 2.1.1. we detail the RNA-Seq experiments comparing the wt strain in the absence and presence of histamine. At lines 160 to 163 we refer to the RNA-Seq experiments comparing the wt strain with the mutant in the HinK transcriptional regulator. The results showed that HinK is the major sensor protein in P. aeruginosa causing transcriptional alterations.
Is it known If P. aeruginosa can secrete histamine or if histamine can modulate increased virulence during exacerbation in the lungs of CF patients? Moreover, considering P. aeruginosa as an opportunistic pathogen, where in the host histamine is secreted which might attract and modulate its virulence? Maybe a subsection of paragraph 3 could be included presenting where histamine has been identified in the host.
Response: What this reviewer mentions is a highly interesting issue. Histamine secretion by P. aeruginosa would have multiple consequences such as a modulation of the immune system, altering P. aeruginosa gene expression and chemoattraction of other P. aeruginosa cells. However, we were unable to localize data on histamine secretion by P. aeruginosa within the host. Maybe this review serves as an inspiration to others to conduct the corresponding studies.
The authors might include a figure similar to Figure 1 specific for E. coli.
Response: E. coli is unable to synthesize and metabolize histamine. It does not have an HDC gene nor the genes identified in P. putida U and P. aeruginosa PAO1 that are required for histamine uptake, metabolism and transcriptional regulation.